# Lowering PyTorch's Memory Consumption for Selective Differentiation

**Samarth Bhatia** [1]   **Felix Dangel** [2]

## Abstract

Memory is a limiting resource for many deep learning tasks. Beside the neural network weights, one main memory consumer is the computation graph built up by automatic differentiation (AD) for backpropagation. We observe that PyTorch's current AD implementation sometimes neglects information about parameter differentiability when storing the computation graph. This information is useful though to reduce memory whenever gradients are requested for a parameter subset, as is the case in many modern fine-tuning tasks. Specifically, inputs to layers that act linearly in their parameters and inputs (fully-connected, convolution, or batch normalization layers in evaluation mode) can be discarded whenever the parameters are marked as non-differentiable. We provide a drop-in, differentiability-agnostic implementation of such layers[1] and demonstrate its ability to reduce memory without affecting run time on popular convolution- and attention-based architectures.

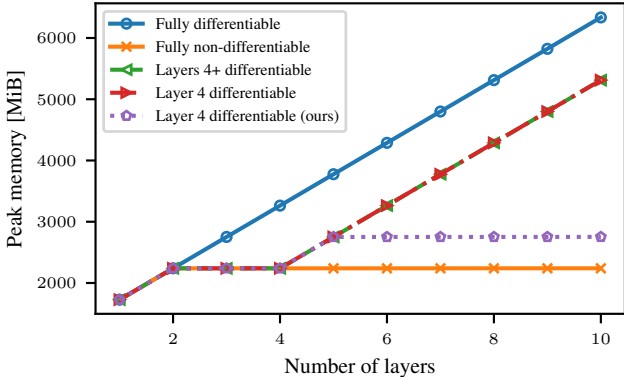

Figure 1: *PyTorch's AD is sometimes not agnostic to parameter differentiability.* We consider a deep CNN made of size-preserving convolutions and measure the forward pass's peak memory when processing a mini-batch of size (256, 8, 256, 256), requiring 512 MiB memory. Memory increases linearly in the number of layers when all parameters are marked differentiable and remains constant when all parameters are marked non-differentiable. Surprisingly, *when only one layer's parameters are marked as differentiable the memory increases as if all subsequent parameters were marked differentiable*. Our drop-in solution stores layer inputs depending on parameter differentiability and reduces memory compared to the current PyTorch implementation.

## 1. Introduction & Motivation

The success of many deep learning applications is driven by scaling computational resources (Thompson et al., 2020). One important resource is GPU memory, specifically on low- and mid-end GPUs which usually offer between 6 to 16 GiB. Therefore, down-scaling the computational demands of deep learning is an important objective to widen its accessibility to researchers and practitioners with fewer hardware resources. Two major memory consumers are the network weights, and the computation graph stored by the automatic differentiation (AD) engine. There exist various

approaches to reduce their memory burden; e.g., the parameters can be compressed with low-precision data types (quantization (Hubara et al., 2018; Li et al., 2017; Nagel et al., 2021)) or sparsified (Hassibi & Stork, 1992; Frantar & Alistarh, 2022), and the computation graph can be off-loaded to CPU (Ren et al., 2021), compressed (Chen et al., 2021), randomized (Oktay et al., 2021), or partially recorded and re-computed (gradient checkpointing (Griewank & Walther, 2008; Chen et al., 2016)).

Here, we consider a special AD scenario we call *selective differentiation* where gradients are requested only for a subset of the computation graph leafs (the neural network inputs and parameters). This approach has gained a lot of practical relevance in the era of large foundation models. For instance, fine-tuning techniques for pre-trained models rely on training only a subset of layers (Zhao et al., 2024; Lee et al., 2023; Zhu et al., 2024; Pan et al., 2024), or extend

---

[1]IIT Delhi [2]Vector Institute. Correspondence to: Samarth Bhatia < samarth.bhatia23@alumni.iitd.ac.in >, Felix Dangel < fdangel@vectorinstitute.ai >.

Accepted to the Workshop on Advancing Neural Network Training at International Conference on Machine Learning (WANT@ICML 2024).

[1]Code and experiments available at github.com/plutonium-239/memsave_torch.

them with parameter-efficient adapters (Hu et al., 2022) which are then trained, keeping the original weights fixed. But selective differentiation is also common in other applications such as generating adversarial examples (Goodfellow et al., 2015) and neural style transfer (Gatys et al., 2016) which optimize the input to a network with frozen weights.

We find that PyTorch's (Paszke et al., 2019) AD allows for additional, simple, optimizations to further reduce memory consumption in the context of selective differentiation:

1. We observe that PyTorch's AD sometimes neglects the differentiability of layer parameters when storing the computation graph (Figure 1). This information is useful though as it allows discarding inputs to linear layers whose parameters are marked as non-differentiable.

2. We provide a drop-in implementation of various layers that is agnostic to the parameter differentiability and demonstrate on various convolutional neural networks (CNNs) and attention-based large language models (LLMs) that it lowers the default implementation's memory footprint without increasing run time.

This easy-to-use insight benefits many tasks with selective differentiation and enables them to scale further. We hope it will stimulate future research into AD optimizations.

## 2. Selective Differentiation in PyTorch

PyTorch users can specify whether they want to compute gradients w.r.t. a tensor through its `requires_grad` attribute, which is dominantly inherited by child tensors: If any input (parent) to an operation is marked differentiable, the output (child) will also be differentiable and require the computation graph to be stored for backpropagation. As we will see, PyTorch does not always check the differentiability of all parents, but rather stores the computation graph as if all parents were differentiable once it encounters a differentiable input. In the following example, we demonstrate this behavior, and how it misses out on possible optimizations for linear operations with non-differentiable parameters.

**Experimental procedure:** We use PyTorch 2.2.1 and measure the forward pass's peak memory of different neural networks as a proxy for the computation graph size stored by the AD engine. We choose peak memory as it is a reasonable proxy for the computation graph size and the relevant metric for causing out-of-memory errors in practise. On GPU, we measure peak memory using `torch.cuda.max_memory_allocated()`, the maximum memory allocated by CUDA. To assess run time differences of our implementation, we compare a forward and backward pass with PyTorch. Each measurement is

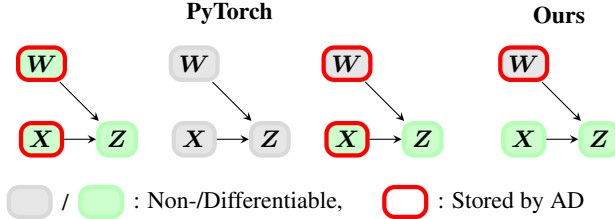

Figure 2: *PyTorch's behaviour of storing the computation graph*, illustrated on a convolution $Z = W * X$. PyTorch stores the layer input whenever it is differentiable, although this is not necessary if the weight does not require gradients. Our approach uses this information to discard the layer input if possible. See Appendix C for computation graphs visualized with `torchviz` (Zagoruyko et al., 2019).

performed in a separate Python session to avoid memory (de)allocation leaks between consecutive runs.

### 2.1. A Simple Example

We start with a synthetic example to probe the internals of PyTorch's AD that are responsible for identifying and storing the computation graph during a forward pass (summarized in Figure 2). We consider a deep CNN consisting entirely of convolutions without bias. Each convolution preserves its input size (kernel size 3, unit padding and stride) and number of channels and we vary its depth as well as the parameter differentiability. As input, we choose a mini-batch of size (256, 8, 256, 256). Storing this tensor, and each intermediate output generated by a layer, requires 512 MiB in single precision. Memory consumed by the convolution kernels of shape (8,8,3,3) is negligible compared to the hidden features. Figure 1 summarizes our findings.

First, we investigate the computation graph size when marking all or no parameters as differentiable. When all parameters are differentiable, all layer inputs must be stored to compute gradients. Consequently, we observe a linear relation with a slope corresponding to the 512 MiB consumed by each intermediate input (Figure 1, *fully differentiable*). When no parameter is differentiable, the memory consumption flattens after more than two layers (Figure 1, *fully non-differentiable*). This is because during a forward pass both input and output tensors of a layer are allocated in memory, in addition to the network's input. Hence, at most two tensors are in memory at a time for a single layer, while at most three are allocated for two or more layers.

Next, we observe the memory consumption in the context of selective differentiation. We consider two scenarios: In the first, all parameters after and including the fourth layer are differentiable, hence all layer inputs after the third layer must be stored in memory. In the second, only the fourth layer is differentiable and therefore only that layer's input is

|  | Memory [GiB] | | | |
| Case | All | Input | Norm | Surgical |
|---|---|---|---|---|
| Default ResNet-101 | **7.83 (1.00)** | 7.83 (1.00) | 7.83 (1.00) | 7.83 (1.00) |
| + swap Convolution | **7.83 (1.00)** | 7.78 (0.99) | 7.78 (0.99) | 7.83 (1.00) |
| + swap ReLU | 8.56 (1.09) | **5.24 (0.67)** | **5.24 (0.67)** | **6.86 (0.88)** |

|  | Time [s] | | | |
| Case | All | Input | Norm | Surgical |
|---|---|---|---|---|
| Default ResNet-101 | 0.45 (1.00) | 0.37 (0.82) | 0.37 (0.82) | 0.41 (0.89) |
| + swap Convolution | 0.45 (0.98) | 0.37 (0.80) | 0.36 (0.80) | 0.40 (0.87) |
| + swap ReLU | 0.44 (0.97) | 0.36 (0.79) | 0.36 (0.78) | 0.41 (0.90) |

| `Bottleneck` | (64, 56, 56) |
|---|---|
| — `Conv2d` | (64, 56, 56) |
| — `BatchNorm2d` | (64, 56, 56) |
| — `ReLU` | (64, 56, 56) |
| — `Conv2d` | (64, 56, 56) |
| — `BatchNorm2d` | (64, 56, 56) |
| — `ReLU` | (64, 56, 56) |
| — `Conv2d` | (64, 56, 56) |
| — `BatchNorm2d` | (256, 56, 56) |
| — `Conv2d` | (64, 56, 56) |
| — `BatchNorm2d` | (256, 56, 56) |
| — `ReLU` | (256, 56, 56) |

Table 1: *GPU peak memory and run time comparison between PyTorch and our memory-saving layers* for ResNet-101. Normalized values are relative to 'All' with PyTorch's default layers. We first swap only the convolution layers for our memory saving alternatives, which does not save memory (see main text for explanation). Then, we swap out other layers like ReLU, which significantly improves memory.

Table 2: *ResNet101's residual block* with layer input shapes, excluding batch size. The output of a ReLU is stored by both the activation layer and the convolution layer it feeds into.

necessary to compute gradients. However, we observe that both scenarios exhibit the *same* memory footprint (Figure 1, *layers 4+ differentiable* and *layer 4 differentiable*)!

We conclude that PyTorch stores a convolution layer's input with `requires_grad = True` although the layer's parameters might be non-differentiable. This information can be useful to reduce the information stored for backpropagation, as we show with our implementation (Figure 1, *layer 4 differentiable (ours)*).

In Appendix B, we repeat similar experiments with fully-connected, transpose convolution, and batch normalization layers. We find that PyTorch's (transpose) convolutions and batch normalization layers in evaluation mode are not agnostic to the differentiability of their weights. Interestingly, fully-connected layers are already optimized.

### 2.2. Implementation Details

Our implementation of memory-saving layers is straightforward and does not require low-level code. For each layer we create a new `torch.autograd.Function` AD primitive, and its associated `torch.nn.Module` layer. The primitive uses the same forward and backward routines as the original operation (from `torch.nn.functional` and `torch.ops.aten`), but has additional logic for the information that is saved to the AD tape, which we describe below. Hence, our implementation shares the performance of PyTorch's. We also provide a converter that replaces supported layers of a net with our memory-saving equivalents.

#### 2.2.1. CONVOLUTION LAYERS

Consider a convolution layer $Z = W * X + b$ with input $X$, output $Z$, kernel $W$, and bias $b$. Its input Jacobian depends on $W$, the weight Jacobian on $X$, and the bias

Jacobian has no dependency (e.g. Dangel, 2023, Chapter 2.3). During the forward pass, we check the differentiability of $W, X$, and only store tensors required by the Jacobians that will be applied during backpropagation. The same dependency pattern holds for other layers that process their inputs linearly w.r.t. their weight and input, and add a bias term, such as (transpose) convolution (Chellapilla et al., 2006) and batch normalization (Ioffe & Szegedy, 2015) layers in evaluation mode. We implement them in exactly the same fashion.

#### 2.2.2. INTERACTION WITH OTHER LAYERS

Real neural nets contain additional layers that are interleaved with linear layers: activations, dropout, pooling, etc. The information stored by such layers may overlap with the input to a linear layer, e.g. if an activation layer feeds into a convolution. Depending on the implementation details, our described optimizations might not apply in such cases because the linear layer's input tensor could still be stored by the preceding activation. We encountered this effect for ReLU layers in real-world CNNs (Section 3), but were able to overcome it using a customized ReLU implementation that saves a boolean mask of the output. As PyTorch currently only supports 1-byte booleans, this leads to a 4x reduction of the stored tensor's size, which could be further reduced to 32x (1-bit booleans are in the works). For dropout layers, we accept adding a slight computational overhead to avoid storing the output by saving the random state and re-generating the mask during backpropagation.

This underlines an important challenge for saving memory in selective differentiation whenever multiple layers use the same tensor for backpropagation. Inputs to linear layers, even if their parameters are non-differentiable, might still be stored by neighboring layers. We believe it should often

| Case | Memory [GiB] | | | | Time [s] | | | |
|---|---|---|---|---|---|---|---|---|
| | **All** | **Input** | **Norm** | **Surgical** | **All** | **Input** | **Norm** | **Surgical** |
| DeepLabv3 (RN101) (Chen et al., 2017) | **22.82 (1.00)** | 22.82 (1.00) | 22.82 (1.00) | 22.82 (1.00) | 0.93 (1.00) | 0.73 (0.79) | 0.73 (0.79) | 0.76 (0.82) |
| + MemSave | 24.90 (1.09) | **15.17 (0.66)** | **15.17 (0.66)** | **16.83 (0.74)** | 0.94 (1.01) | 0.76 (0.82) | 0.76 (0.81) | 0.79 (0.85) |
| EfficientNetv2-L (Tan & Le, 2019; 2021) | **26.81 (1.00)** | 26.81 (1.00) | 26.81 (1.00) | 26.81 (1.00) | 0.77 (1.00) | 0.62 (0.81) | 0.62 (0.81) | 0.68 (0.88) |
| + MemSave | 26.81 (1.00) | **18.64 (0.70)** | **18.64 (0.70)** | **22.05 (0.82)** | 0.78 (1.02) | 0.63 (0.82) | 0.63 (0.82) | 0.69 (0.90) |
| FCN (RN101) (Long et al., 2015) | **22.23 (1.00)** | 22.23 (1.00) | 22.23 (1.00) | 22.23 (1.00) | 0.83 (1.00) | 0.67 (0.80) | 0.67 (0.80) | 0.70 (0.84) |
| + MemSave | 24.39 (1.10) | **15.15 (0.68)** | **15.15 (0.68)** | **16.80 (0.76)** | 0.87 (1.04) | 0.70 (0.84) | 0.69 (0.83) | 0.74 (0.88) |
| Faster-RCNN (RN101) (Ren et al., 2015) | **6.84 (1.00)** | 6.84 (1.00) | 6.84 (1.00) | 6.84 (1.00) | 0.77 (1.00) | 0.66 (0.86) | 0.66 (0.85) | 0.69 (0.89) |
| + MemSave | 7.31 (1.07) | **4.79 (0.70)** | **4.79 (0.70)** | **5.73 (0.84)** | 0.77 (0.99) | 0.65 (0.84) | 0.67 (0.86) | 0.68 (0.88) |
| MobileNetv3-L (Howard et al., 2019) | **2.82 (1.00)** | 2.82 (1.00) | 2.82 (1.00) | 2.82 (1.00) | 0.39 (1.00) | 0.32 (0.82) | 0.32 (0.82) | 0.35 (0.89) |
| + MemSave | 2.96 (1.05) | **1.91 (0.68)** | **1.91 (0.68)** | **2.52 (0.89)** | 0.40 (1.01) | 0.32 (0.82) | 0.32 (0.82) | 0.35 (0.89) |
| ResNeXt101-64x4d (Xie et al., 2017) | **15.15 (1.00)** | 15.15 (1.00) | 15.15 (1.00) | 15.15 (1.00) | 0.65 (1.00) | 0.53 (0.82) | 0.53 (0.82) | 0.58 (0.90) |
| + MemSave | 16.75 (1.11) | **9.87 (0.65)** | **9.87 (0.65)** | **13.32 (0.88)** | 0.64 (0.98) | 0.52 (0.80) | 0.52 (0.80) | 0.56 (0.87) |
| SSDLite (MobileNetv3-L) (Sandler et al., 2018) | **0.54 (1.00)** | 0.53 (0.97) | 0.53 (0.97) | 0.53 (0.97) | 0.63 (1.00) | 0.59 (0.92) | 0.55 (0.87) | 0.58 (0.91) |
| + MemSave | 0.57 (1.04) | **0.41 (0.75)** | **0.41 (0.75)** | **0.50 (0.92)** | 0.66 (1.05) | 0.57 (0.90) | 0.55 (0.86) | 0.58 (0.92) |
| VGG-16 (Simonyan & Zisserman, 2015) | 4.93 (1.00) | 4.93 (1.00) | N/A | 5.05 (1.02) | 0.37 (1.00) | 0.28 (0.77) | N/A | 0.31 (0.84) |
| + MemSave | **4.30 (0.87)** | **3.08 (0.62)** | N/A | **3.15 (0.63)** | 0.38 (1.05) | 0.30 (0.82) | N/A | 0.33 (0.89) |

Table 3: *GPU peak memory and run time comparison between PyTorch and our memory-saving layers* for CNNs.

be possible to resolve this scenario—albeit through careful implementation—e.g. whenever the Jacobian can be implemented by either storing the layer input or output, as is the case for various activation functions, and pooling layers.

## 3. Real-World Examples

Here we measure the effect of our memory-saving layers on real-world CNNs. We consider four different scenarios:

**All:** All network parameters are differentiable. This serves as reference to establish similar performance of our layers in the absence of selective differentiation.

**Input:** Only the input to the neural net is differentiable. This situation resembles constructing adversarial examples (Goodfellow et al., 2015) or style-transfers (Gatys et al., 2016) by optimizing noisy inputs.

**Surgical:** Only the first quarter of layers are differentiable. This situation is similar to surgical fine-tuning (Lee et al., 2023), which splits a network into different blocks, each containing a subset of layers, to be trained one at a time.

**Norm:** Only normalization layers are differentiable, resembling layer norm fine-tuning in LLMs (Zhao et al., 2024).

We report GPU results on an NVIDIA RTX A6000 with 48 GiB of VRAM. All CNNs are fed inputs of size (64, 3, 224, 224) and we measure according to the procedure described in Section 2. For object detection models, the batch size is 4 and 2 boxes are predicted per input image.

### 3.1. ResNet-101

For ResNet-101, we take a detailed look at our memory-saving layer's effects. It is a decently powerful, modern model and frequently used as a backbone for other architectures such as CLIP (Radford et al., 2021) and LAVA (Gurram et al., 2022). It contains dense, convolution, batch normalization, ReLU, and max/average pooling layers. Table 1 summarizes our findings (model in training mode).

**PyTorch is often unaware of selective differentiation:** In the upper part of Table 1, we see that the default PyTorch implementation uses the same amount of memory for any scenario. In fact, marking only the input to a neural net as differentiable consumes as much memory as marking all parameters, although the former does not require storing the inputs to fully-connected and convolution layers. This confirms the findings on the toy model from Section 2.

**Layer interactions diminish memory savings:** To gradually investigate the effect of our layers, we only swap out the convolutions; *without* observing any effects. At first, this seems counter-intuitive. However, a closer look at the architecture (Table 2) reveals that the convolutions are preceded by ReLU activations which unconditionally store their outputs and render our convolution layer's optimizations ineffective as described in Section 2.2.2.

After swapping out ReLU with our mask-based custom implementation, we observe substantial memory savings. E.g., memory consumption for 'Input' dropped to roughly two thirds. Only when all parameters are differentiable, we see a slight increase in memory after swapping out ReLUs. This is to be expected as the mask stored by our custom implementation requires additional storage.

**Run time remains unaffected:** The bottom half of Table 1 shows that run time for a fixed case remains equal up to measurement noise. This confirms that our memory-saving layers share the default implementation's performance.

| Case | Memory [GiB] | | | | Time [s] | | | |
|---|---|---|---|---|---|---|---|---|
| | All | Input | Norm | Surgical | All | Input | Norm | Surgical |
| **SiLU Transformers** | | | | | | | | |
| LLaMa3-8B (Touvron et al., 2023) ($H = 4096$) | 31.01 (1.00) | 27.27 (0.88) | 28.26 (0.91) | 28.18 (0.91) | 1.39 (1.00) | 1.13 (0.81) | 1.13 (0.82) | 1.17 (0.84) |
| + MemSave | **29.01 (0.94)** | **26.26 (0.85)** | **26.26 (0.85)** | **26.94 (0.87)** | 1.61 (1.16) | 1.04 (0.75) | 1.05 (0.75) | 1.12 (0.81) |
| Mistral-7B (Jiang et al., 2023) ($H = 4096$) | 41.67 (1.00) | 34.20 (0.82) | 36.17 (0.87) | 36.01 (0.86) | 2.09 (1.00) | 1.55 (0.74) | 1.56 (0.75) | 1.67 (0.80) |
| + MemSave | **37.67 (0.90)** | **32.17 (0.77)** | **32.17 (0.77)** | **33.54 (0.80)** | 2.55 (1.22) | 1.44 (0.69) | 1.46 (0.70) | 1.71 (0.82) |
| Phi3-4B (Gunasekar et al., 2023) ($H = 3072$) | 31.74 (1.00) | 26.01 (0.82) | 27.49 (0.87) | 27.40 (0.86) | 1.59 (1.00) | 1.23 (0.78) | 1.24 (0.78) | 1.31 (0.83) |
| + MemSave | **28.74 (0.91)** | **24.49 (0.77)** | **24.49 (0.77)** | **25.55 (0.81)** | 1.69 (1.06) | 1.09 (0.69) | 1.12 (0.71) | 1.23 (0.78) |
| **ReLU Transformers** | | | | | | | | |
| Transformer (Vaswani et al., 2017) ($H = 2048$) | 26.91 (1.00) | 21.54 (0.80) | 21.54 (0.80) | 23.04 (0.86) | 2.47 (1.00) | 2.21 (0.90) | 2.19 (0.89) | 2.24 (0.91) |
| + MemSave | **25.60 (0.95)** | **20.23 (0.75)** | **20.23 (0.75)** | **21.73 (0.81)** | 2.57 (1.04) | 2.26 (0.92) | 2.25 (0.91) | 2.30 (0.93) |
| T5 (Raffel et al., 2020) ($H = 768$) | 33.40 (1.00) | 25.94 (0.78) | 28.85 (0.86) | 27.77 (0.83) | 1.70 (1.00) | 1.37 (0.81) | 1.39 (0.82) | 1.47 (0.86) |
| + MemSave | **31.84 (0.95)** | **22.80 (0.68)** | **22.80 (0.68)** | **25.28 (0.76)** | 1.95 (1.14) | 1.52 (0.89) | 1.54 (0.90) | 1.59 (0.94) |

Table 4: *GPU peak memory and run time comparison between PyTorch and our memory-saving layers* for LLMs.

## 3.2. Results on CNNs

To further solidify our findings, we now evaluate our layers on other popular and commonly used CNNs, including ResNet-18 (He et al., 2016), VGG-16 (Simonyan & Zisserman, 2015), ResNeXt101-64x4d (Xie et al., 2017), EfficientNetv2-L (Tan & Le, 2019; 2021), MobileNetv3-L (Sandler et al., 2018; Howard et al., 2019), FCN with a ResNet-101 backbone (Long et al., 2015), DeepLabv3 with a ResNet-101 backbone (Chen et al., 2017), Faster-RCNN with a ResNet-50 backbone (Ren et al., 2015), and SSDLite with a MobileNetv3-L backbone (Sandler et al., 2018). Table 3 summarizes the comparison.

On this large repertoire of networks, we observe the same effects as on ResNet-101 from the previous section: Due to our customized ReLU implementation, memory is slightly higher in the absence of selective differentiation. Run time is unaffected by swapping in our layers, while the selective differentiation scenarios consistently show lower memory consumption (as low as two thirds), underlining the usefulness of our approach in this context.

So far, we used all batch normalization layers in training mode. In Appendix A, we experiment with `BatchNorm2d` in evaluation mode. This allows discarding normalization layer inputs whenever the parameters are marked non-differentiable and enables further memory savings, with reductions up to 6x and no runtime overhead (Table 5).

## 3.3. Results on Transformers

Selective differentiation is of high practical relevance for large language models (LLMs), especially with the increasing popularity of vision language models (VLMs) and the release of models such as LLaVa, PaLI, PaLI-Gemma and GPT-4o (Liu et al., 2023; Chen et al., 2023b;a). Another area where this is quite important is using a (light) modality-specific encoder to encode (and project) the input of different modalities into the embedding space of the LLM, such

as in scientific ML (Shen et al., 2024).

Here, we investigate the impact of our memory-saving layers on attention-based models. Popular libraries like Hugging-Face that provide access to such models often implement attention through linear layers, which we did not find to suffer from the behavior of convolutions. However, an additional challenge in these architectures is dropout, which may interact with inputs to linear layers similarly to other activations (Section 2.2.2). Thus, we find our tricks for ReLU and dropout helpful to avoid storing inputs to linear layers if their weights are marked non-differentiable.

We consider the same cases as in Section 3 and all nets consume embeddings of size (64, 256, $H$), with hidden size $H$ of a network (given in Table 4). We test on the vanilla Transformer (Vaswani et al., 2017), T5 (Raffel et al., 2020), LLaMa3 (Touvron et al., 2023), Mistral (Jiang et al., 2023) and Phi3 (Abdin et al., 2024; Gunasekar et al., 2023). The results are shown in Table 4. With LLaMa (8B parameters), Mistral (7B parameters) and Phi3 (4B parameters) being large models, they are loaded in `bfloat16` data type and the batch size is reduced to 8 for LLaMa and 16 for the others. We observe that our approach also enables memory savings on attention-based models and consistently achieves a smaller memory footprint.

## 4. Conclusion

We have shown how to improve the memory consumption of PyTorch's automatic differentiation in the context of selective differentiation where gradients are only requested for a subset of variables—a common situation in modern fine-tuning tasks. Our approach is based on the insight that PyTorch sometimes ignores the differentiability of tensors; specifically in (transpose) convolutions and batch normalization layers in evaluation mode. To overcome this, we provide a drop-in implementation which takes into account the differentiability of all tensors when storing the computation graph. Empirically, we demonstrated the effectiveness

of our approach to reduce memory in multiple selective differentiation cases without affecting run time, on both convolution- and attention-based architectures. Our method is easy to use, requiring only a single call to a converter function that replaces all supported layers with our equivalents. Next, we plan to study its impact on real-world applications, such as parameter-efficient LLM fine-tuning with low-rank adapters (Hu et al., 2022).

## Acknowledgements

Resources used in preparing this research were provided, in part, by the Province of Ontario, the Government of Canada through CIFAR, and companies sponsoring Vector Institute.

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

## A. Results on CNNs when `BatchNorm` layers are set to `eval` mode

| | Memory [GiB] | | | | Time [s] | | | |
|---|---|---|---|---|---|---|---|---|
| case | All | Input | Norm | SurgicalFirst | All | Input | Norm | SurgicalFirst |
| DeepLabv3 (RN101) (Chen et al., 2017) | **22.82 (1.00)** | 22.82 (1.00) | 22.82 (1.00) | 22.82 (1.00) | 0.88 (1.00) | 0.66 (0.75) | 0.70 (0.79) | 0.70 (0.79) |
| + MemSave | 24.90 (1.09) | **4.28 (0.19)** | **15.17 (0.66)** | **8.40 (0.37)** | 0.90 (1.02) | 0.69 (0.79) | 0.72 (0.82) | 0.73 (0.83) |
| EfficientNetv2-L (Tan & Le, 2019; 2021) | **26.81 (1.00)** | 26.81 (1.00) | 26.81 (1.00) | 26.81 (1.00) | 0.74 (1.00) | 0.59 (0.79) | 0.59 (0.80) | 0.64 (0.86) |
| + MemSave | 26.81 (1.00) | **10.45 (0.39)** | **18.64 (0.70)** | **17.31 (0.65)** | 0.75 (1.01) | 0.58 (0.78) | 0.59 (0.80) | 0.66 (0.89) |
| FCN (RN101) (Long et al., 2015) | **22.23 (1.00)** | 22.23 (1.00) | 22.23 (1.00) | 22.23 (1.00) | 0.79 (1.00) | 0.59 (0.75) | 0.62 (0.79) | 0.63 (0.80) |
| + MemSave | 24.39 (1.10) | **4.26 (0.19)** | **15.15 (0.68)** | **7.99 (0.36)** | 0.81 (1.03) | 0.63 (0.79) | 0.65 (0.82) | 0.67 (0.85) |
| Faster-RCNN (RN101) (Ren et al., 2015) | **6.84 (1.00)** | 6.84 (1.00) | 6.84 (1.00) | 6.84 (1.00) | 0.75 (1.00) | 0.64 (0.84) | 0.64 (0.85) | 0.67 (0.88) |
| + MemSave | 7.31 (1.07) | **1.98 (0.29)** | **4.79 (0.70)** | **4.19 (0.61)** | 0.74 (0.99) | 0.63 (0.83) | 0.63 (0.83) | 0.66 (0.87) |
| MobileNetv3-L (Howard et al., 2019) | **2.82 (1.00)** | 2.82 (1.00) | 2.82 (1.00) | 2.82 (1.00) | 0.40 (1.00) | 0.31 (0.78) | 0.31 (0.78) | 0.34 (0.86) |
| + MemSave | 2.96 (1.05) | **0.87 (0.31)** | **1.91 (0.68)** | **2.10 (0.75)** | 0.39 (0.98) | 0.31 (0.79) | 0.31 (0.79) | 0.35 (0.87) |
| ResNeXt101-64x4d (Xie et al., 2017) | **15.15 (1.00)** | 15.15 (1.00) | 15.15 (1.00) | 15.15 (1.00) | 0.62 (1.00) | 0.48 (0.77) | 0.50 (0.80) | 0.53 (0.85) |
| + MemSave | 16.75 (1.11) | **2.46 (0.16)** | **9.87 (0.65)** | **9.77 (0.64)** | 0.61 (0.99) | 0.47 (0.77) | 0.49 (0.79) | 0.52 (0.84) |
| SSDLite (MobileNetv3-L) (Sandler et al., 2018) | **0.54 (1.00)** | 0.53 (0.97) | 0.53 (0.97) | 0.53 (0.97) | 0.62 (1.00) | 0.56 (0.90) | 0.52 (0.84) | 0.57 (0.91) |
| + MemSave | 0.57 (1.04) | **0.26 (0.48)** | **0.41 (0.75)** | **0.44 (0.82)** | 0.63 (1.02) | 0.58 (0.93) | 0.53 (0.86) | 0.57 (0.92) |

Table 5: *GPU peak memory and run time comparison between PyTorch and our memory-saving layers* for CNNs with `BatchNorm` layers in `eval` mode. VGG-16 has been excluded here as it does not contain any `BatchNorm` layers.

## B. Probing PyTorch Layers in the Presence of Selective Differentiation

Our experiments with a deep CNN of size-preserving convolutions from Figure 1 and Section 2.1 revealed that PyTorch's 2d convolution layer stores its input tensor whenever it is differentiable, irrespective of the weight's differentiability. Here, we perform analogous experiments, but with other layers we suspect to exhibit similar behaviour. Specifically, we consider layers whose forward pass is linear w.r.t. both the layer input and the layer weights. This includes fully-connected layers (`torch.nn.Linear`), convolution layers (`torch.nn.ConvNd`), transpose convolution layers (`torch.nn.ConvTransposeNd`), and batch normalization layers in evaluation mode (e.g. `torch.nn.BatchNorm2d`).

Layers are set up to preserve their input size and we feed mini-batches that require $512\,\text{MiB}$ storage in single precision:

- $(512, 1024, 256)$ for `torch.nn.Linear`
- $(4096, 8, 4096)$ for `torch.nn.Conv1d` and `torch.nn.ConvTranspose1d`
- $(512, 8, 256, 256)$ for `torch.nn.BatchNorm2d`, `torch.nn.Conv2d`, and `torch.nn.ConvTranspose2d`
- $(64, 8, 64, 64, 64)$ for `torch.nn.Conv3d` and `torch.nn.ConvTranspose3d`

Compared to the input and intermediate activations, the memory footprint of a layer's weight is negligible:

- $(256, 256)$ for `torch.nn.Linear`
- $(8, 8, 3)$ for `torch.nn.Conv1d` and `torch.nn.ConvTranspose1d`
- $(8)$ for `torch.nn.BatchNorm2d`
- $(8, 8, 3, 3)$ for `torch.nn.Conv2d` and `torch.nn.ConvTranspose2d`
- $(8, 8, 3, 3, 3)$ for `torch.nn.Conv3d` and `torch.nn.ConvTranspose3d`

Our results are summarized in Figure 3. We can see that PyTorch's convolutions, transpose convolutions, and batch normalization (in evaluation mode) are not agnostic to the differentiability of their weights. PyTorch's fully-connected layer, however, is already agnostic to the differentiability of its weights.

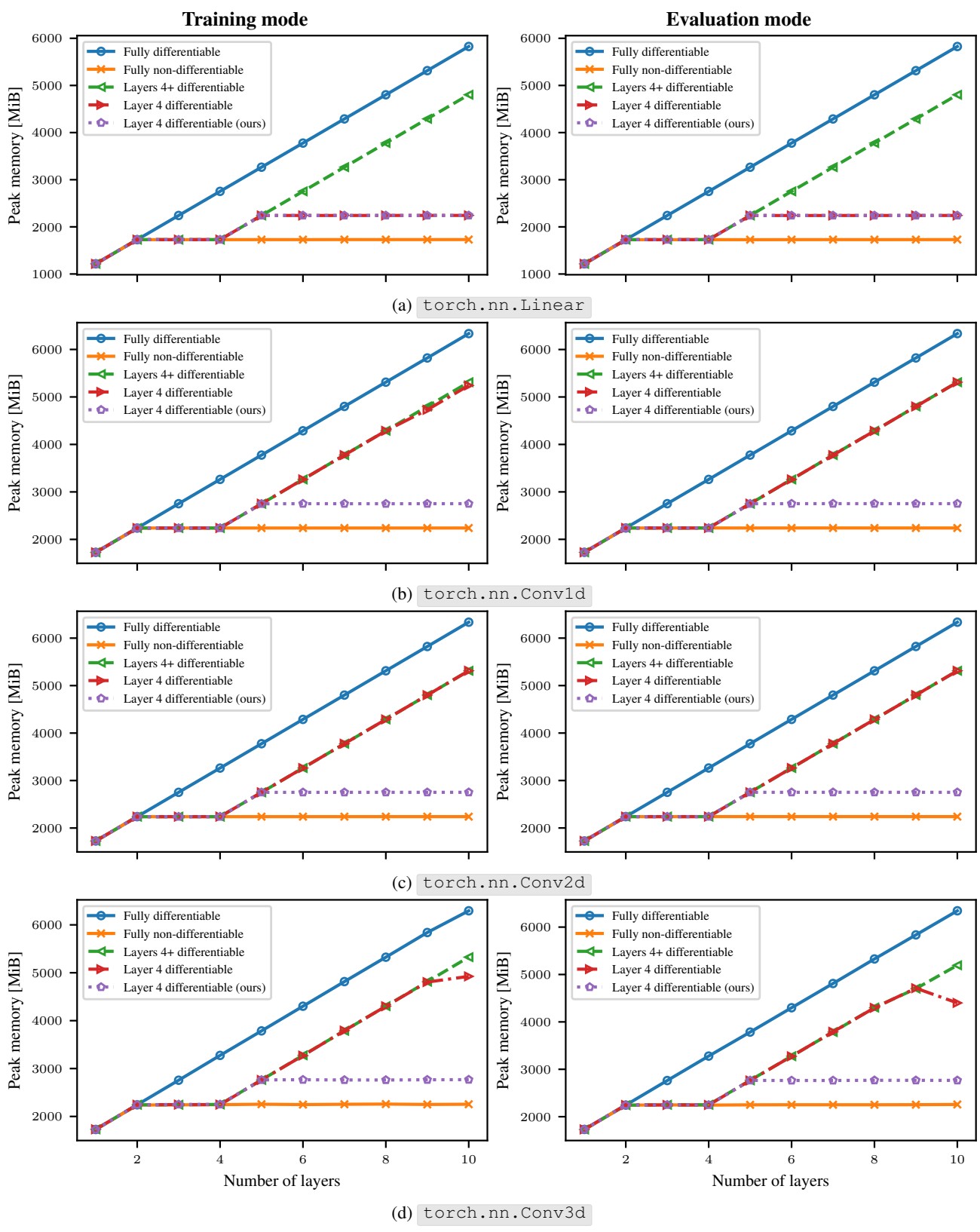

(a) `torch.nn.Linear`

(b) `torch.nn.Conv1d`

(c) `torch.nn.Conv2d`

(d) `torch.nn.Conv3d`

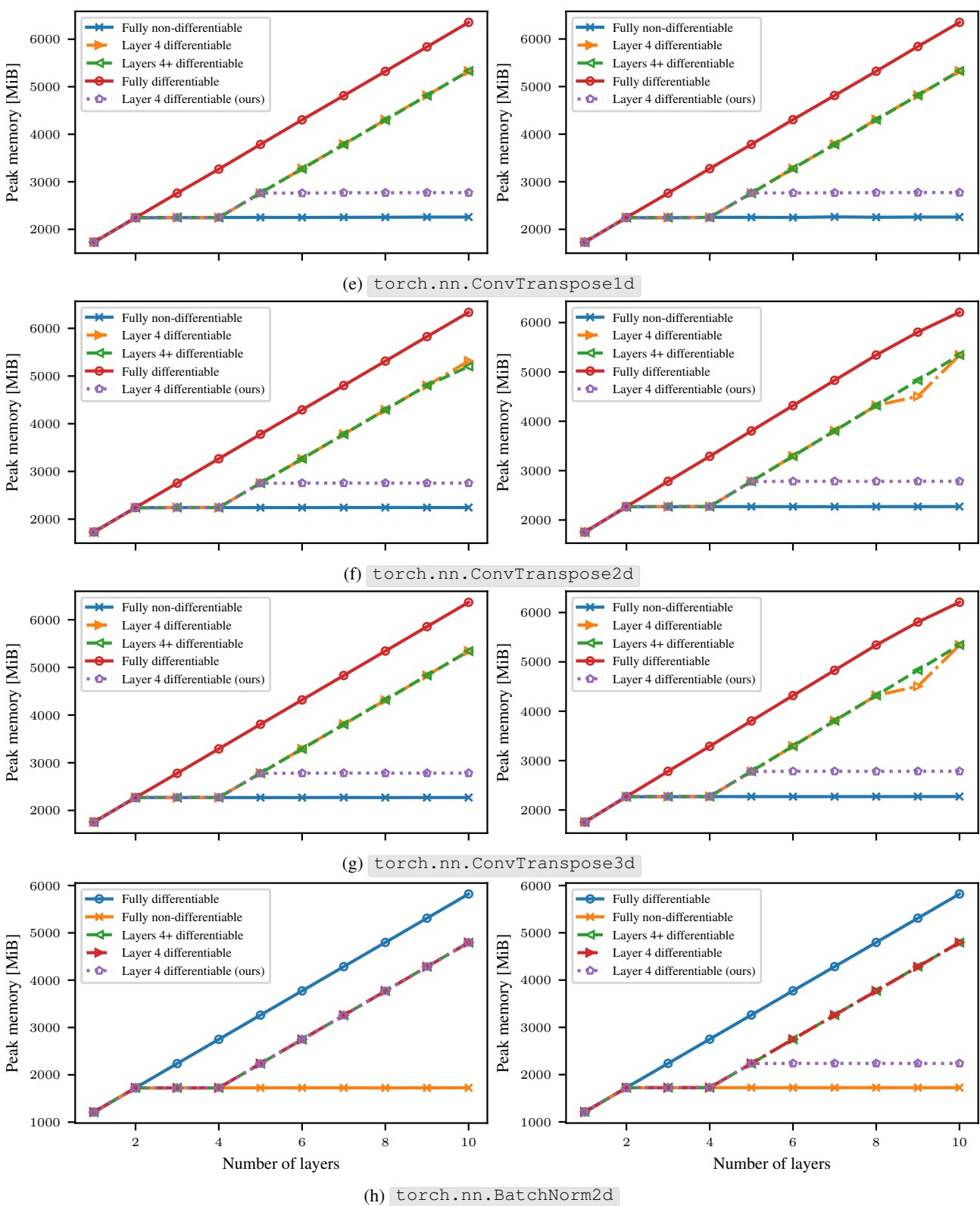

(e) `torch.nn.ConvTranspose1d`

(f) `torch.nn.ConvTranspose2d`

(g) `torch.nn.ConvTranspose3d`

(h) `torch.nn.BatchNorm2d`

Figure 3: *Probing different PyTorch layer's for their awareness of parameter differentiability.* (a) PyTorch's `nn.Linear` is agnostic to parameter differentiability. (b, c, d) PyTorch's `nn.ConvNd`, (e, f, g) `nn.ConvTransposeNd`, and (h) `nn.BatchNorm2d` in evaluation mode are not agnostic to parameter differentiability.

## C. Diagrams showing the computation graph for elementary layers

In all the following figures, we show PyTorch's behavior of saving tensors when they are not required. For all layers, the **Input** case is shown.

The colors mean the following (taken from the `torchviz` package (Zagoruyko et al., 2019)):

**blue** A node representing the main operation being discussed

**green** A node representing any tensor

**gray** A node representing any other operations (i.e. `View` / `AccumulateGrad` etc.)

Tensors which are saved by the AD engine are marked as `[saved tensor]`, and they also have an undirected edge to the main operation node.

### C.1. `Conv2d`

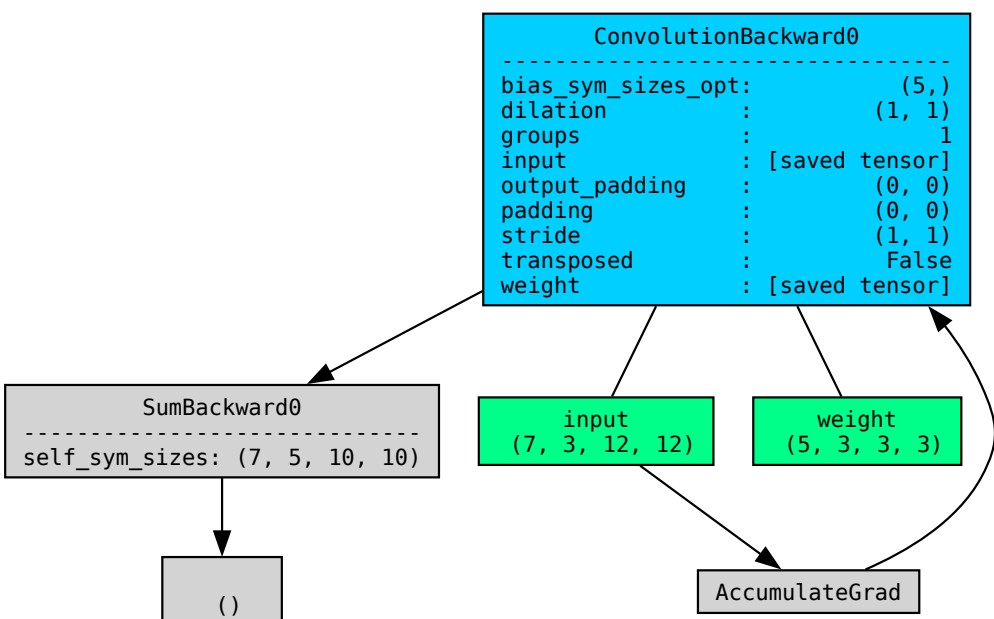

PyTorch Conv2d - **Input** Case
Differentiable: Weights ×, Input ✓

Figure 4: Computation graphs of a convolution layer for the **Input** case. Even though the input is differentiable, PyTorch saves it (as can be seen inside the `ConvolutionBackward0` node - the input is of shape `[7, 3, 12, 12]`). MemSave on the other hand, does not save the input.

## C.2. `Linear`

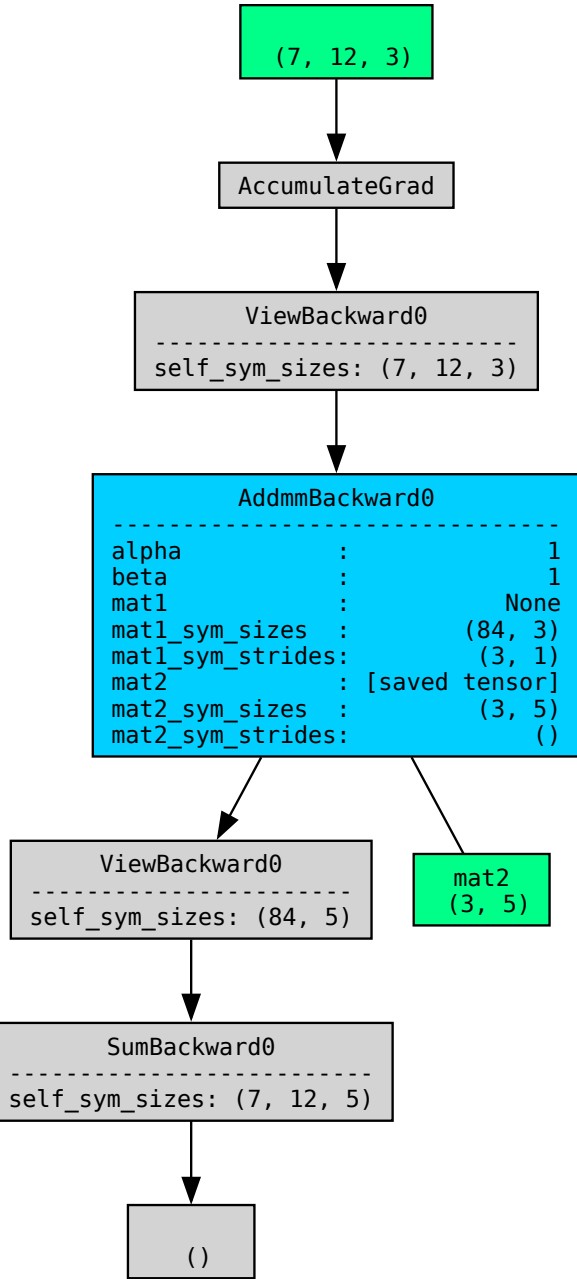

PyTorch Linear - **Input** Case
Differentiable: Weights $\times$, Input $\checkmark$

Figure 5: Computation graphs of $y = Wx + b$ for the **Input** case. This visualization is interesting because these graphs indicate that a linear layer does not save its input if the weight is marked non-differentiable, which is already the optimal behaviour.

## C.3. `BatchNorm2d`

**Training Mode**

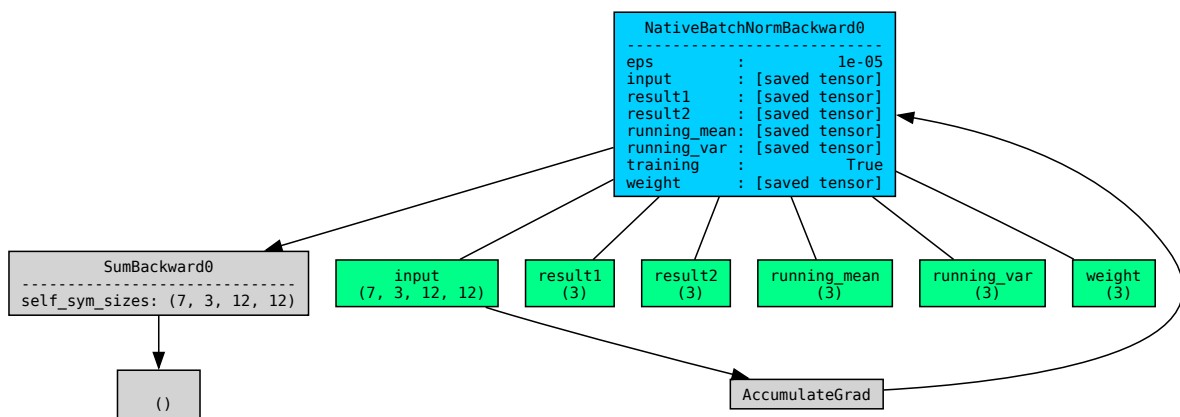

PyTorch BatchNorm - **Input** Case
Differentiable: Weights $\times$ Input $\checkmark$

Figure 6: Computation graphs of $y = W \frac{x - \mu(x)}{\sqrt{\sigma^2(x) + \epsilon}} + b$ (i.e., BatchNorm in training mode) for the **Input** case.

**Evaluation Mode**

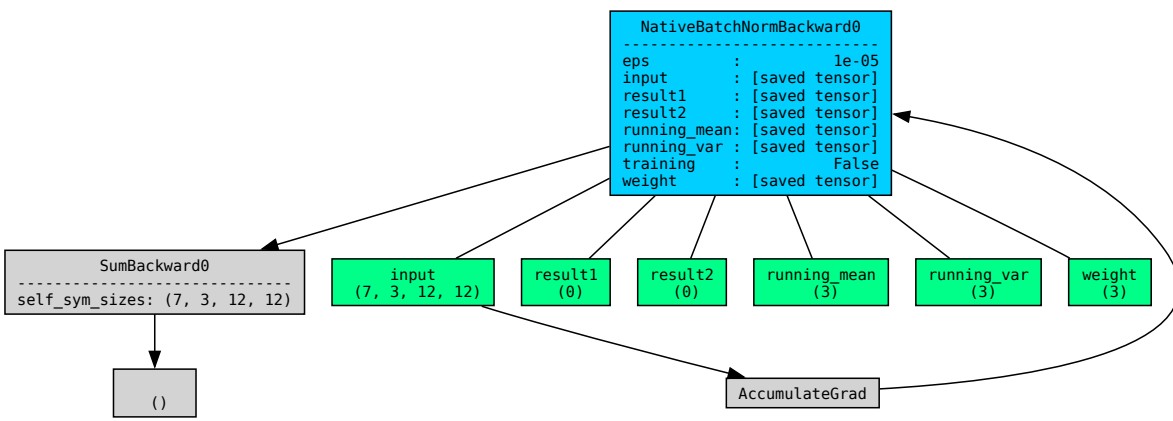

PyTorch BatchNorm (eval mode) - **Input** Case
Differentiable: Weights $\times$ Input $\checkmark$

Figure 7: Computation graphs of $y = W \frac{x - \hat{\mu}}{\sqrt{\hat{\sigma}^2 + \epsilon}} + b$ (i.e., BatchNorm in eval mode) for the **Input** case. PyTorch also saves the input, even though it is not required for calculating the input gradient. MemSave recognizes this and does not save the input.

