# OpenReview forum: "Lowering PyTorch's Memory Consumption for Selective Differentiation"
_ICML.cc/2024/Workshop/WANT — WANT@ICML 2024 Poster_

### Official Review · Reviewer_x1T2 · 2024-06-07
**Good Idea but Maybe Too Simple**

**Confidence:** 4

**Summary:**

Authors present a simple optimization of memory usage in scenarios where not all parameters are required to be differentiable (e.g. finetuning, adapters, etc.). Idea is clear and simple. They rely on a specific version of a specific framework (PyTorch), which limits its usage. Authors didn't show how recent Torch's compilers affects memory usage. I leave it to workshop organizers to decide if this paper should be present there

**Strengths:**

* Very clear and simple optimization idea

**Weaknesses:**

* After PyTorch major version 2, I would like to see how just-in-time compiler affects this behavior and hence memory usage

* (Philosophical) Authors treat PyTorch as a black box, they conduct experiments to check memory consumption and how autodiff works, though its code is open sourced, it's not a black box overall

**Limitations:**

It only works with PyTorch of specific version, for the current moment (June 7, 2024) the version used in the paper is outdated, so potentially the autodiff's behavior shown in the paper could have changed already or might change in future.

**Suggestions:**

It would be nice to see if this behavior persists in other frameworks and if it changes after compilation

---

### Official Review · Reviewer_PzSW · 2024-06-13
**Encouraging preliminary experiments, but still insufficiently formalized regarding relevant and useful observation**

**Confidence:** 4

**Summary:**

This paper presents a novel technique for saving memory for PyTorch activations in the case where some layers are non-differentiable.

**Strengths:**

The initial observation about PyTorch's behavior is interesting. It would have been interesting to extend it to other frameworks to check if this is a PyTorch specific feature.

**Weaknesses:**

In my opinion, the paper lacks a more formal analysis to assess when descendants should inherit the differentiable character. This more formal study would be a first step towards more automatic detection (using compilation tools) and would help to strengthen the findings.

**Limitations:**

Finally, the proposed solution lacks generality and requires rewriting certain layers and models, and is still limited in scope (e.g., for normalization layers).

**Suggestions:**

In conclusion, this is an interesting and original paper that starts from an observation about PyTorch's behavior that is useful in practice for limiting memory consumption in many interesting contexts (adversarial examples, fine-tuning,...). There's still a lot of work to be done to formalize, generalize and automate, but it's an interesting paper for the WS audience, with a convincing set of preliminary experiments.

---

### Meta-Review · Area_Chair_kjsK · 2024-06-16

**Recommendation:** Accept (Poster)
**Confidence:** 3

**Metareview:**

## Strengths
* This paper describes a very clear idea, that does allow to obtain better memory consumption
* Experiments are convincing
## Weaknesses
* It is not clear how general the results of the paper can be.
* The experiments are very preliminary and require more generalization and automation
* Some of the insight could be found by reading the code rather than experimenting

The general sentiment about this paper is that the idea is interesting, despite the results being rather preliminary. I recommend
acceptance as a poster.

---

### Decision · Program_Chairs · 2024-06-17

**Decision:**

Accept (Poster)

**Comment:**

We thank the authors for their time and contribution to WANT and we are pleased to share that after the reviewing process the paper has been accepted. Congratulations! We encourage the authors to consider reviewers' feedback for the improvement of the camera-ready version. We hope to see you in person at the workshop and brainstorm on efficient training research together!